# Hepatitis E Virus Infection Caused Elevation of Alanine Aminotransferase Levels in a Patient with Chronic Hepatitis B and Choledocholithiasis

**DOI:** 10.3390/reports6040055

**Published:** 2023-11-17

**Authors:** Rei Hirano, Tatsuo Kanda, Masayuki Honda, Shuhei Arima, Mai Totsuka, Ryota Masuzaki, Shini Kanezawa, Reina Sasaki-Tanaka, Naoki Matsumoto, Hiroaki Yamagami, Tomotaka Ishii, Masahiro Ogawa, Shuzo Nomura, Mariko Fujisawa, Kei Saito, Masaharu Takahashi, Hiroaki Okamoto, Hirofumi Kogure

**Affiliations:** 1Division of Gastroenterology and Hepatology, Department of Medicine, Nihon University School of Medicine, 30-1 Oyaguchi-kamicho, Itabashi-ku, Tokyo 173-8610, Japan; 2Division of Virology, Department of Infection and Immunity, Jichi Medical University School of Medicine, 3311-1 Yakushiji, Shimotsuke, Tochigi 329-0498, Japanhokamoto@jichi.ac.jp (H.O.)

**Keywords:** common bile duct stones, HBV, HEV, prolonged jaundice

## Abstract

Hepatitis E virus (HEV) genotypes 3 and 4 are zoonotic strains that are primarily transmitted through the consumption of undercooked pork or game meat. They also cause asymptomatic infections, acute hepatitis, acute-on-chronic liver failure, chronic hepatitis, and extrahepatic manifestations. Here, we report a man in his 80s who had chronic hepatitis B, took entecavir for it, and presented with higher levels of alanine aminotransferase (ALT) and jaundice. An abdominal computed tomography scan revealed choledocholithiasis with cholecystolithiasis. Although endoscopic papillary balloon dilatation was performed for the removal of a common bile duct stone, the abnormal liver function tests, including jaundice, were prolonged. After other viral hepatitis and other causes of the liver injury were ruled out, as his serum was positive for immunoglobulin A anti-HEV and HEV genotype 3b RNA, we diagnosed him as having acute hepatitis E. In this case, with chronic hepatitis B and a common bile duct stone, the prolonged abnormal results for the liver function tests seemed to be caused by HEV infection. In conclusion, in cases with high ALT levels after removing choledocholithiasis, other factors, including HEV infection, should be considered to determine the cause of abnormal liver function test results. The further examination of hepatitis D virus infection and high ALT levels may be needed in HBV-infected individuals.

## 1. Introduction

Prolonged cholestasis is a rare complication associated with endoscopic retrograde cholangiopancreatography (ERCP) [1]. Prolonged cholestasis and liver injury after ERCP and the removal of common bile duct stones may be associated with contrast agents, estrogen therapy and antibiotics [1,2,3,4].

Higher levels of alanine aminotransferase (ALT) are occasionally caused by viral hepatitis, drug-induced liver injury, choledocholithiasis, and/or circulatory failure [5,6]. These complications may cause the postponement of discharge in certain cases.

Hepatitis E virus (HEV) infection causes asymptomatic infections, acute hepatitis, acute-on-chronic liver failure, chronic hepatitis, and extrahepatic manifestations [7]. HEV genotypes 3 and 4 cause zoonotic infections in humans and can be contracted through the consumption of contaminated pork or game meat, direct contact, or other routes [7]. Infections with HEV genotypes 3 and 4 are often observed in Japan [8].

## 2. Case Presentation

A man in his 80s with a diagnosis of chronic hepatitis B was referred to our hospital with appetite loss, general fatigue, jaundice, nausea, and marked liver dysfunction (Table 1). His medical history was as follows: he underwent appendicitis surgery at a younger age; surgery on the left hydrocele in a testicle 21 years ago; radiation and hormone therapies for prostate cancer for 19 years; partial hepatectomy for hepatocellular carcinoma (HCC) 3 years ago; and transurethral bladder cancer resection 2 years ago. He had no history of transfusion, tattoos, drug abuse, or drinking alcohol and no family history of liver diseases. He had a history of drug allergy to piperacillin sodium.

The patient had taken entecavir (0.5 mg daily) for 15 years at a local clinic, and his liver function has been stable other than the occurrence of HCC 3 years prior. Abnormal liver function tests were observed at regular blood tests one month before. One day before the visit to our hospital, severely abnormal liver function test results were obtained: aspartate aminotransferase (AST), 1272 IU/L; ALT, 958 IU/L; γ-glutamyl transpeptidase (γ-GTP), 197 IU/L; and total bilirubin, 5.2 mg/dL. The presence of common bile duct stones was suspected after abdominal ultrasonography. He was then introduced to our hospital due to his abnormal liver dysfunction test results.

The patient’s body length and body weight were 164 cm and 56.4 kg, respectively. His blood pressure, pulse rate, and body temperature were 125/83 mmHg, 18/min, and 36.1 °C, respectively. He was conscious, his conjunctiva bulbi were icteric, and surgery scars were observed on his soft abdominal surface. Computed tomography images taken upon admission to our hospital demonstrated cholelithiasis in the gallbladder (Figure 1a) and choledocholithiasis in the common bile duct (Figure 1b).

We performed endoscopic papillary balloon dilatation (EPBD) with a 6 mm balloon for the common bile duct stone and removed it using a Medi-Globe 8-Wire Nitinol Basket (Medico’s Hirata, Osaka, Japan), as the exacerbation of his jaundice and abnormal liver function test results were thought to be caused by the common bile duct stone (Figure 2).

However, the patient’s liver function test results did not improve immediately. Approximately 2 weeks after admission, his serum was confirmed to be positive for both immunoglobulin A (IgA) anti-HEV and HEV RNA. The IgG anti-HEV, IgM anti-HEV, and IgA anti-HEV levels were measured using an enzyme-linked immunosorbent assay (ELISA), and HEV RNA was measured via real-time RT-PCR according to previously described methods [9,10,11,12,13]. We diagnosed him as having HEV genotype 3b infection; as his hepatitis B virus (HBV) DNA levels were 1 log IU/mL (LIU/mL), the acute exacerbation or breakthrough of chronic hepatitis B were also excluded (Table 1). When he was discharged from our hospital 16 days after admission, his ALT and total bilirubin levels had improved (57 IU/L and 2.5 mg/dL, respectively), although he was still positive for HEV RNA (Figure 3). Notably, the titer of HEV RNA determined via the previously described method [11] was found to be lower (3.8 × 10^2^ copies/mL) on discharge than it was on days 5, 7, 12, and 14 (4.4 × 10^4^ copies/mL, 2.4 × 10^4^ copies/mL, 2.3 × 10^3^ copies/mL, and 8.5 × 10^2^ copies/mL, respectively). The other causes of viral hepatitis and autoimmune liver diseases were also ruled out. The drug lymphocyte stimulation test results (DLSTs) were also negative for flavoxate hydrochloride, vibegron, and silodosin [14]. Finally, we diagnosed him with an acute HEV infection after he was discharged.

## 3. Discussion

In this case of chronic HBV infection and choledocholithiasis, HEV infection caused the higher ALT levels. As patients with cirrhosis undergoing surgery or ERCP/sphincterotomy may have a high mortality rate from bleeding, EPBD could be used in these patients [15]. Instead of endoscopic sphincterotomy (EST), EPBD was selected to treat a common bile duct stone because our patient had a history of hepatectomy for HCC, and the operator thought it would be better to preserve the function of the sphincter in the case of the recurrence of HCC, as the annual recurrence rates of HCC have been increasing [16].

According to the World Health Organization (WHO), the incubation period following exposure to HEV ranges from 2 to 10 weeks (with mean ranges from 5 to 6 weeks), and HEV-infected persons excrete HEV beginning from a few days before to 3–4 weeks after the onset of the disease [17]. They also stated that chronic HEV genotype 3 or 4 infections had been reported in immunosuppressed people, particularly organ transplant recipients on immunosuppressive drugs. Unfortunately, we could not evaluate the HEV RNA and ALT levels 2–3 months after the documented episode to confirm a negative HEV RNA result or further reduction in the ALT levels; although, upon following up with this patient after 16 days, we measured a decrease in both the HEV genotype 3b RNA levels and ALT levels with jaundice.

Bjornsson et al. [5] reported that no patients with HEV infection were identified among 142 adult patients with ALT levels of more than 500 IU/L from an annual total of 24,193 patients who came to the University Hospital in Iceland. A seroepidemiological study demonstrated the low prevalence of hepatitis E in Iceland [18]. Only 6 (2.1%) patients tested positive for IgG anti-HEV among 291 persons, including 195 healthy volunteers, 21 pig farm workers, and 75 patients participating in a nationwide study on drug-induced liver injury [18].

Mechnik et al. [19] reported a 70-year-old male in Israel with cholelithiasis in the gallbladder and a total bilirubin level of 25 mg/dL, which were caused by HEV infection. There is a report about the significant presence of autochthonous acute HEV infections in Israel [20]. The high HEV seropositivity of the farm workers, together with the previous identification of HEV genotype 3 in human sewage, suggests circulation among the humans in Israel [21]. Thus, although some patients may present elevated ALT levels due to HEV infection, the prevalence could depend on the epidemic situation of HEV infection in their country [21,22].

Although pronounced ALT elevation is a part of the clinical symptoms of common bile duct stones, especially among patients with a smaller bile duct diameter [23]; compared with the patients with common bile duct stones and drug-induced liver injury, the patients with HEV infection had a significantly higher maximum ALT and ALT/alkaline phosphatase (ALP) ratio [24]. Wallace et al. [24] suggested that the patients with ALT levels equal to or higher than 300 IU/L should be tested for HEV.

HEV genotypes 1 and 2 are clearly responsible for human hepatitis, and they are spread through the fecal–oral route via contaminated water in Asia and Africa and in Africa and Mexico, respectively [7]. The incidence of autochthonous HEV infections (predominantly genotypes 3 and 4) is increasing in industrialized countries such as Japan [25,26,27]. The present case supports the previous reports that HEV genotype 3, as well as other genotypes, can induce acute-on-chronic liver failure (ACLF) in patients with underlying liver diseases [28,29,30].

Ribavirin may be considered for patients with severe acute hepatitis E or ACLF [7]. Ribavirin may be one effective therapeutic agent for HEV-induced ACLF [31]. Unfortunately, there is no specific therapy for pregnant women with HEV-induced ACLF [7]. Additionally, it may be difficult for HEV-infected patients with renal failure or severe anemia to use ribavirin [32]. Therefore, novel antiviral drugs should be developed for HEV infection [33,34,35].

Azithromycin and ritonavir strongly inhibited HEV production in culture supernatants [33]. According to the results from in silico and in vitro screening, amodiaquine and lumefantrine may be good antimalarial drug candidates for repurposing against HEV [36]. The combination of ribavirin and sofosbuvir may be effective for autochthonous acute HEV infection [37]. The inhibitors of heat shock protein 90 (HSP90) (iHSP90) markedly suppressed HEV replication [38,39]. A potential drug-like inhibitor against HEV-Methyltransferase, 3-(4-Hydroxyphenyl)propionic acid (HPPA), is an effective HEV inhibitor [40]. Li et al. [41] reported that the anti-cancer drug gemcitabine inhibited HEV through inducing an interferon-like response via the activation of STAT1 phosphorylation. Human liver-derived organoids [42,43] and microRNA signatures in HEV infection [44,45] could be helpful for the development of antivirals against HEV infection.

The HEV genome, with a size of ~7200 nucleotides, encodes at least three open reading flames (ORFs), and the ORF2 protein is translated into the capsid protein (~600 amino acid (aa) residues) [46]. Zhang et al. [46] identified 144 conserved sites among the 660 aa residues. ORF2 encodes the capsid protein, which elicits neutralizing antibodies [47].

HEV p495 manufactured by GSK was used as a vaccine antigen, and it showed good safety and efficacy in a phase II clinical trial. HEV p495-genotype 1 (aa 112–608) expressed by Baculovirus was used as a vaccine antigen and showed good safety in a phase II clinical trial [48,49,50,51]. HEV p239-genotype 1 (aa 368–606) expressed by *E. coli*, with a trade name of Hecolin^®^, was licensed in China in 2011 [48,52,53,54]. Thus, the development of anti-HEV drugs and HEV vaccines is a very important issue [55].

It has been reported that hepatitis A virus (HAV) infection is associated with acute acalculous cholecystitis in children and adults [56,57]. In these cases, higher elevations of the ALT levels of more than 500 IU/L are occasionally observed [58,59,60,61,62,63,64,65,66,67,68,69]. Cholestasis is likely to be associated with HAV infection [57]. HBV and hepatitis C virus (HCV) infections also cause acute acalculous cholecystitis [70,71], occasionally leading to higher elevations of the ALT levels of more than 500 IU/L [71]. Fujioka et al. [72] reported that HEV genotype 1 infection led to acute acalculous cholecystitis as an extrahepatic manifestation and a higher elevation of the ALT levels of 2842 IU/L. Piza Palacios et al. [73] also reported that a co-infection with HAV and HEV caused acute acalculous cholecystitis as an extrahepatic manifestation. Hepatitis virus infection may impair the blood flow of the hepatobiliary tract as well as the liver. It is important to distinguish acute acalculous cholecystitis in HEV-infected cases when higher elevations of the ALT levels are observed.

Choledocholithiasis, cholelithiasis, HBV, being older, male, having diabetes, smoking, drinking alcohol, and obesity are potential risk factors for cholangiocarcinoma development [74,75]. It is unknown whether HBV infection is associated with choledocholithiasis [76]. A significantly increased risk of developing gallstones among HCV-infected patients was reported [77,78,79,80,81,82,83]. HEV genotype 3 infection was also found to be associated with gallstone-related diseases, including biliary pancreatitis [84].

This patient had HBV-related cirrhosis with a history of HCC. A history of liver conditions was also associated with an excess of biliary stones [85]. The prevalence of common bile duct stones is three times higher in patients with cirrhosis than it is in those without cirrhosis [86,87,88]. EST is a safer option for the management of choledocholithiasis in patients with cirrhosis [89]. There is an overall higher rate of adverse events related to ERCP in patients with cirrhosis, especially hemorrhages and post-ERCP pancreatitis [90]. EPBD without sphincterotomy is a safer and more effective treatment for choledocholithiasis in patients with cirrhosis [15].

In this case, hepatitis D virus (HDV) RNA was not measured. The prevalence of HDV infection is low in Japan, seen in 4 (1.2%) out of 323 HBsAg carriers in the Tokyo–Chiba area [91], although it was found in 32 (8.5%) of 375 HBsAg-positive patients in Miyako Island, Okinawa [92]. Although HBV DNA was well suppressed before, during and after the elevation of the ALT levels, the involvement of HDV infection may not be completely ruled out. In general, HDV infection can be successfully blocked using a combination of hepatitis B immunoglobulin and a vaccine [92]. 

The recent recommendations [93] make the most sense in the countries with a low prevalence of HBV and those with a high prevalence of HBV and HDV. The recently measured prevalence of HDV infection in patients with HBV infection in Hokkaido, Japan, was 1.7% [94]. The patients with sera positive for the anti-HDV antibody experienced the rapid progression of liver fibrosis, suggesting the importance of routine HDV testing. As the number of people migrating from abroad into Japan increases, the prevalence of HDV infection may increase. Further studies are needed at this point [95,96,97,98,99,100,101].

In Japan, the national health insurance system approved the use of the IgA anti-HEV antibody for the diagnosis of hepatitis E. Takahashi et al. [10] reported that, in solid-phase ELISA, the IgA anti-HEV assay is significantly more specific than the IgM anti-HEV assay with regard to the ability to diagnose hepatitis E; IgA anti-HEV could be a first-choice marker as diagnostic indicator of recent HEV infection; and the diagnostic accuracy increases when the positive results obtained from the IgA anti-HEV assay are confirmed via the additional or simultaneous detection of IgM anti-HEV.

## 4. Conclusions

We reported a male patient with chronic hepatitis B and choledocholithiasis, who presented with a high elevation of ALT levels, which was caused by HEV infection. It is important to consider the other factors, including HEV infection, in patients with prolonged abnormal liver function after the removal of common bile duct stones. Further investigation of the role of HDV infection in the elevation of ALT levels is also needed.

## Figures and Tables

**Figure 1 reports-06-00055-f001:**
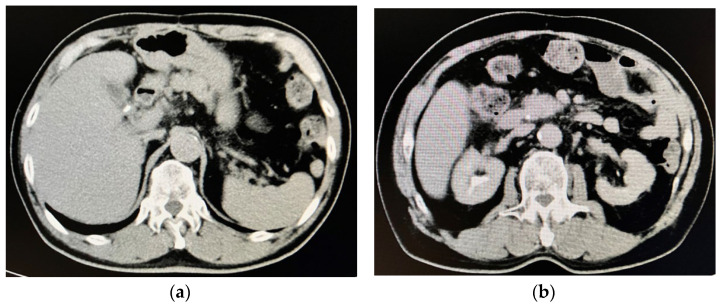
Computed tomography images on admission to our hospital. (**a**) Cholelithiasis was observed in the gallbladder. (**b**) Choledocholithiasis was observed in the common bile duct.

**Figure 2 reports-06-00055-f002:**
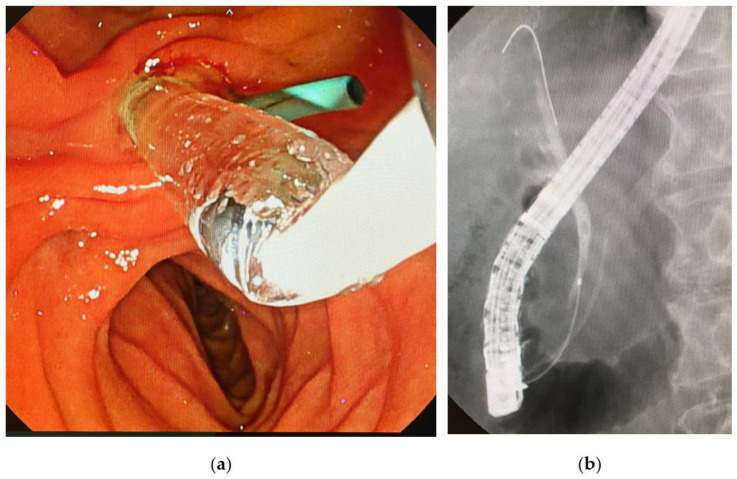
Endoscopic papillary balloon dilatation (EPBD) for a bile duct stone in this case. (**a**) EPBD for the management of the common bile duct stone. The sky-blue-colored tube is a prophylactic pancreatic stent. (**b**) Cholangiogram showing a 5 mm stone in the common bile duct.

**Figure 3 reports-06-00055-f003:**
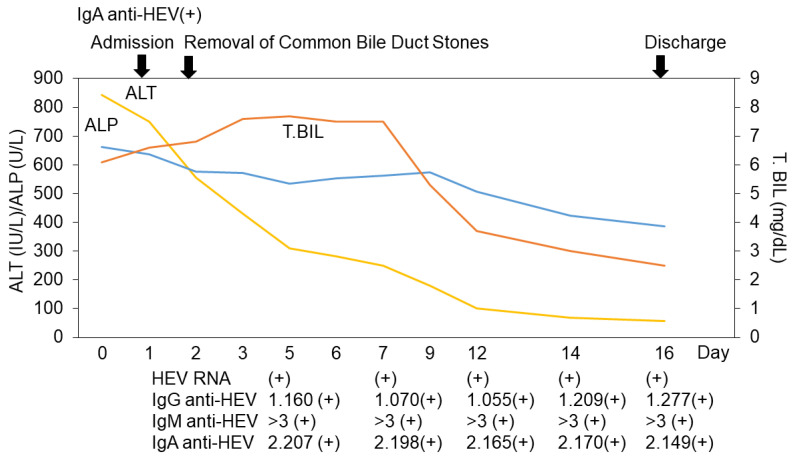
Clinical course of this case. ALT, alanine aminotransferase; ALP, alkaline phosphatase; T. Bil, total bilirubin; HEV, hepatitis E virus; Ig, immunoglobulin. IgG anti-HEV, IgM anti-HEV, and IgA anti-HEV levels were measured using an enzyme-linked immunosorbent assay (ELISA), and HEV RNA was measured via real-time RT-PCR according to previously described methods [9,10,11,12,13].

**Table 1 reports-06-00055-t001:** Laboratory data on first admission.

Item	Value	Item	Value	Item	Value
WBC	4400/µL	AST	964 IU/L	Amylase	276 IU/L
Hemoglobin	11.3 g/dL	ALT	843 IU/L	HBsAg	Positive
Platelets	136,000/µL	LDH	282 IU/L	Anti-HBc	Positive
Neutrophils	48%	ALP	662 IU/L	IgM anti-HBc	Negative
Eosinophils	1.0%	γ-GTP	172 IU/L	HBV DNA	1.0 LIU/mL
Monocytes	18%	CK	80 U/mL	Anti-HCV	Negative
Lymphocytes	29%	T. Bil	6.1 mg/dL	IgM anti-HAV	Negative
Atypical Lymphocytes	4.0%	D. Bil	5.0 mg/dL	IgA anti-HEV	Positive
PT	100%	TP	7.6 g/dL	HEV RNA	Positive
PT-INR	0.92	Albumin	3.4 g/dL	HEV Genotype	3b
Glucose	94 mg/dL	T. CHO	141 mg/dL	Anti-HIV	Negative
HbA1c	6.4%	TG	207 mg/dL	ANA	80-fold
AFP	1.9 ng/mL	BUN	27 mg/dL	IgG	2243 mg/dL
PIVKA-II	27 mAU/mL	Creatinine	1.0 mg/dL	IgA	481 mg/dL
SARS-CoV-2 antigen	Negative	CRP	2.33 mg/dL	IgM	265 mg/dL

WBC, white blood cell count; PT, prothrombin time; PT-INR, PT international normalized ratio; HbA1c, hemoglobin A1c; AFP, α-fetoprotein; PIVKA-II, protein induced by vitamin K absence or antagonist-II; SARS-CoV-2 antigen, severe acute respiratory syndrome coronavirus 2 antigen; AST, aspartate aminotransferase; ALT, alanine aminotransferase; LDH, lactate dehydrogenase; ALP, alkaline phosphatase; γ-GTP, γ-glutamyl transpeptidase; CK, creatine kinase; T. Bil, total bilirubin; D. Bil, direct bilirubin; TP, total protein; T. CHO, total cholesterol; TG, triglyceride; BUN, blood urea nitrogen; CRP, C-reactive protein; HBsAg, hepatitis B virus (HBV) surface antigen; anti-HBc, anti-HBV core antibody; IgM anti-HBc, immunoglobulin M anti-HBc; anti-HCV, anti-hepatitis C virus (HCV) antibody; IgM anti-HAV, immunoglobulin M anti-hepatitis A virus; IgA anti-HEV, immunoglobulin A anti-hepatitis E virus; anti-HIV, anti-human immunodeficiency virus antibody; ANA, anti-nuclear antibody. IgG anti-HEV, IgM anti-HEV, and IgA anti-HEV levels were measured using an enzyme-linked immunosorbent assay (ELISA), and HEV RNA was measured via a real-time RT-PCR according to previously described methods [9,10,11,12,13].

## Data Availability

The data underlying this study are available in this article.

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
