# Peer review of "Hepatitis E Virus Infection Caused Elevation of Alanine Aminotransferase Levels in a Patient with Chronic Hepatitis B and Choledocholithiasis"

_reports, 2023, doi:10.3390/reports6040055_

Round 1

Reviewer 1 Report

Comments and Suggestions for Authors

            The authors report a case of a patient with HBV infection who presented with ALT elevation during antiviral treatment. Serological analysis showed the presence of IgA antibodies against HEV and HEV-RNA positivity by RT-PCR. The information is interesting, although some concerns should be addressed before acceptance of this manuscript.

1.     Abstract: presented with…

2.     The authors show a follow up of 16 days of the patient. During all this follow up, the patient exhibited positivity to HEV RNA, as expected for an acute infection. According to WHO and literature, viremia can be found for up to 7 seeks in acute cases, and chronic infection are only found in immunosuppressed individuals and infected with genotypes 3 or 4. It would be interesting to complement this study with an evaluation of transaminases and HEV RNA at 2-3 months of the documented episode, to confirm a negative of HEV RNA result at present, and reduction of the hepatic enzymes levels.

3.     In Case presentation, the date of the diagnosis should be included.

4.     The Methods are not described: brand of serological assays, HEV-RT-PCR reference, etc… at least as legends in table and figures.

5.     Why were tested IgA antibodies and not IgM?

6.     Another important serological marker that should have been excluded is HDV coinfection.

7.     Discussion, lines 114-155: the authors should rephrase this statement as suggested here: HEV seemed to be responsible for the elevation of ALT observed in the patient. And again, absence of HEV RNA in a follow up (2-3 months later) with a reduction of ALT would support their observation and association with HEV.

8.     Discussion, lines 148- 163 and 164-173: the information described there is not directly related to the Case report. The authors should edit this information and focus more on the topic of this study. The number of references cited related to this section might also be reduced.

9.     Conclusions: HDV should also be mentioned, particularly because of an underlying HBV infection. It would be advisable, as stated before, to include a HDV serology in this study, and mention it also in Discussion and Conclusions.

Comments on the Quality of English Language

Some minor edition: in Abstract, presented with

Author Response

Response to the comments from Reviewer 1:

Thank you very much for your invaluable comments.

Reply to your comment 1: “Abstract: presented with…”

Thank you very much for your invaluable comments. We agree with you. We revised our manuscript accordingly.

Reply to comment 2: “The authors show a follow up of 16 days of the patient. During all this follow up, the patient exhibited positivity to HEV RNA, as expected for an acute infection. According to WHO and literature, viremia can be found for up to 7 seeks in acute cases, and chronic infection are only found in immunosuppressed individuals and infected with genotypes 3 or 4. It would be interesting to complement this study with an evaluation of transaminases and HEV RNA at 2-3 months of the documented episode, to confirm a negative of HEV RNA result at present, and reduction of the hepatic enzymes levels.”

Thank you very much for your invaluable comments. We agree with you. We revised our manuscript accordingly as follows.

In Discussion section, lines 129-138,

…as the annual recurrence rates of HCC have been increasing [16].

According to the World Health Organization (WHO), the incubation period following exposure to HEV ranges from 2 to 10 weeks (with mean ranges of 5 to 6 weeks), and HEV-infected persons excrete HEV beginning from a few days before to 3–4 weeks after the onset of the disease [17]. They also stated that chronic HEV genotype 3 or 4 infections had been reported in immunosuppressed people, particularly organ transplant recipients on immunosuppressive drugs. Unfortunately, we could not evaluate HEV RNA and ALT levels 2–3 months after the documented episode to confirm a negative HEV RNA result or further reduction in ALT levels, although, upon following up with this patient after 16 days, we measured a decrease in both HEV genotype 3b RNA levels and ALT levels with jaundice.

Reply to your comment 3: “In Case presentation, the date of the diagnosis should be included.…”

Thank you very much for your invaluable comments. We agree with you. However, we could not describe the date of the diagnosis because of the personal information. We revised our manuscript accordingly as follows.

In Case Presentation section, line 114s-115,

…vibegron, and silodosin [14]. Finally, we diagnosed him with an acute HEV infection after he was discharged.

Reply to your comment 4: “The Methods are not described: brand of serological assays, HEV-RT-PCR reference, etc… at least as legends in table and figures.…”

Thank you very much for your invaluable comments. We agree with you. We revised Table 1 legend and Figure 3 legend of the revised manuscript accordingly. We also revised our manuscript as follows.

In Case Presentation section, lines 99-104,

…liver function tests did not improve immediately. Approximately 2 weeks after admission, his serum was confirmed to be positive for both immunoglobulin A (IgA) anti-HEV and HEV RNA. IgG anti-HEV, IgM anti-HEV, and IgA anti-HEV were measured using an enzyme-linked immunosorbent assay (ELISA), and HEV RNA was measured via real-time RT-PCR, according to previously described methods 9–13]. We diagnosed him as having an HEV genotype 3b infection;…

Reply to your comment 5: “Why were tested IgA antibodies and not IgM?”

Thank you very much for your invaluable comments. We agree with you. We revised our manuscript as follows.

In Discussion section, lines 234-240,

In Japan, the national health insurance system approved the use of the IgA anti-HEV antibody for the diagnosis of hepatitis E. Takahashi et al. [10] reported that, in solid-phase ELISA, the IgA anti-HEV assay is significantly more specific than the IgM anti-HEV assay with regard to the ability to diagnose hepatitis E; that IgA anti-HEV could be a first-choice marker as diagnostic indicator of recent HEV infection; and that the diagnostic accuracy increases when positive results obtained from the IgA anti-HEV assay are confirmed via the additional or simultaneous detection of IgM anti-HEV.

Reply to your comment 6: “Another important serological marker that should have been excluded is HDV coinfection.”

Thank you very much for your invaluable comments. We agree with you. We revised our manuscript as follows.

In Discussion section, lines 220-233,

…in patients with cirrhosis [15].

In the present case, hepatitis D virus (HDV) RNA was not measured. The prevalence of HDV infection is low in Japan, seen in 4 (1.2%) out of 323 HBsAg carriers in the Tokyo–Chiba area [91], although it was found in 32 (8.5%) of 375 HBsAg-positive patients in Miyako Island, Okinawa [9]. Although HBV DNA was well suppressed before, during and after the elevation of ALT levels, the involvement of HDV infection may not be completely ruled out. In general, HDV infection can be successfully blocked using a combination of hepatitis B immunoglobulin and a vaccine [9].

Recent recommendations [93] make the most sense in countries with a low prevalence of HBV and those with a high prevalence of HBV and HDV. The recently measured prevalence of HDV infection in patients with HBV infection in Hokkaido, Japan, was 1.7% [94]. Patients with serum positive for anti-HDV antibody experienced rapid progression of liver fibrosis, suggesting the importance of routine HDV testing. As the number of people migrating from abroad into Japan increases, the prevalence of HDV infection may increase. Further studies are needed at this point.

Reply to your comment 7: “Discussion, lines 114-155: the authors should rephrase this statement as suggested here: HEV seemed to be responsible for the elevation of ALT observed in the patient. And again, absence of HEV RNA in a follow up (2-3 months later) with a reduction of ALT would support their observation and association with HEV.”

Thank you very much for your invaluable comments. We agree with you. We revised our manuscript as follows.

In Discussion section, lines 129-138,

…as the annual recurrence rates of HCC have been increasing [16].

According to the World Health Organization (WHO), the incubation period following exposure to HEV ranges from 2 to 10 weeks (with mean ranges of 5 to 6 weeks), and HEV-infected persons excrete HEV beginning from a few days before to 3–4 weeks after the onset of the disease [17]. They also stated that chronic HEV genotype 3 or 4 infections had been reported in immunosuppressed people, particularly organ transplant recipients on immunosuppressive drugs. Unfortunately, we could not evaluate HEV RNA and ALT levels 2–3 months after the documented episode to confirm a negative HEV RNA result or further reduction in ALT levels, although, upon following up with this patient after 16 days, we measured a decrease in both HEV genotype 3b RNA levels and ALT levels with jaundice.

Reply to your comment 8: “Discussion, lines 148- 163 and 164-173: the information described there is not directly related to the Case report. The authors should edit this information and focus more on the topic of this study. The number of references cited related to this section might also be reduced.”

Thank you very much for your invaluable comments. We disagree with you. This is important part of our manuscript. However, we revised our manuscript as follows.

In Discussion section, lines 167-193,

…Ribavirin may be considered for patients with severe acute hepatitis E or ACLF [7]. Ribavirin may be one effective therapeutic agent for HEV-induced ACLF [31]. Unfortunately, there is no specific therapy for pregnant women with HEV-induced ACLF [7]. Additionally, it may be difficult for HEV-infected patients with renal failure or severe anemia to use ribavirin [32]. Therefore, novel antiviral drugs should be developed for HEV infection [33–35].

Azithromycin and ritonavir strongly inhibited HEV production in culture supernatants [33]. According to results from in silico and in vitro screening, amodiaquine and lumefantrine may be good antimalarial drug candidates for repurposing against HEV [36]. The combination of ribavirin and sofosbuvir may be effective for autochthonous acute HEV infection [37]. Inhibitors of heat shock protein 90 (HSP90) (iHSP90) markedly suppressed HEV replication [38,39]. A potential drug-like inhibitor against HEV-Methyltransferase, 3-(4-Hydroxyphenyl)propionic acid (HPPA), is an effective HEV inhibitor [40]. Li et al. [41] reported that the anti-cancer drug gemcitabine inhibited HEV through inducing an interferon-like response via the activation of STAT1 phosphorylation. Human liver-derived organoids [42,43] and microRNA signatures in HEV infection [44,45] could be helpful for the development of antivirals against HEV infection.

The HEV genome, with a size of ~7200 nucleotides, encodes at least three open reading flames (ORFs), and the ORF2 protein is translated into the capsid protein (~600 amino acid (aa) residues) [46]. Zhang et al. [46] identified 144 conserved sites among the 660 aa residues. ORF2 encodes the capsid protein, which elicits neutralizing antibodies [47].

HEV p495, manufactured by GSK, was used as a vaccine antigen, and it showed good safety and efficacy in a phase II clinical trial. HEV p495-genotype 1 (aa 112–608), expressed by Baculovirus, was used as a vaccine antigen and showed good safety in a phase II clinical trial [48–51]. HEV p239-genotype 1 (aa 368–606), expressed by E.coli, with a trade name of Hecolin®, was licensed in China in 2011 [48, 52–54]. Thus, the development of anti-HEV drugs and HEV vaccines is a very important issue [55].

….

Reply to your comment 9: “Conclusions: HDV should also be mentioned, particularly because of an underlying HBV infection. It would be advisable, as stated before, to include a HDV serology in this study, and mention it also in Discussion and Conclusions.”

Thank you very much for your invaluable comments. We agree with you. We revised our manuscript accordingly as follows.

In Abstract section, lines 33-34,

,,, of abnormal liver function test results. Further examination of hepatitis D virus infection and high ALT levels may be needed in HBV-infected individuals.

In Conclusions section, lines 242-246,

…We reported a male case with chronic hepatitis B and choledocholithiasis, who presented with a high elevation in ALT levels which was caused by HEV infection. It is important to consider other factors, including HEV infection, in patients with prolonged abnormal liver function after the removal of common bile duct stones. Further investigation of the role of HDV infection in the elevation of ALT levels is also needed.

Reviewer 2 Report

Comments and Suggestions for Authors

The case report Hepatitis E Virus Infection Causes Higher Elevation of Alanine Aminotransferase Levels in a Patient with Chronic Hepatitis B and Choledocholithiasis does bring some interesting data but has a few issues with presentation. First, as the patient has been in care for 15 years due to chronic hepatitis B infection, HCC and some other health problems, it would be interesting to compare his lab results before, with emphasis on liver enzymes and perhaps, find a sera that can be tested negative for HEV antibody or RNA to conclude when has the patient been infected with that particular pathogen. If old sera samples are not available, I would like authors to state why has the patient been tested for HEV. Is that a part of the standard pre-operational workup? Or was there a suspicion that he might have contracted it somewhere (other than elevated liver enzyme values)? I would like the authors to debate more other possible causes, other than jus a sentence in the conclusion. There is an abundance of references (probably a bit more than a case report needs, including the some 10% of the papers where one of the authors of this manuscript has been listed), most of those used in the Discussion section. I advise the authors to rewrite the discussion part, putting the emphasis on their case, other possible causes of the elevated aminotransferase and then comparing it to other similar case reports.

Comments on the Quality of English Language

I advise the authors to have the manuscript re-checked by an English language professor or a native speaker, starting from the title itself (saying "higher elevation" is probably not the best way to express the topic of this manuscript).

Author Response

Response to the comments from Reviewer 2:

Thank you very much for your invaluable comments.

Reply to your comment 1: “First, as the patient has been in care for 15 years due to chronic hepatitis B infection, HCC and some other health problems, it would be interesting to compare his lab results before, with emphasis on liver enzymes and perhaps, find a sera that can be tested negative for HEV antibody or RNA to conclude when has the patient been infected with that particular pathogen. If old sera samples are not available, I would like authors to state why has the patient been tested for HEV. Is that a part of the standard pre-operational workup? Or was there a suspicion that he might have contracted it somewhere (other than elevated liver enzyme values)?”

Thank you very much for your invaluable comments. Sure. According to your comments, we revised our manuscript and described the limitation of study more clearly.

In Case Presentation section, lines 99-115,

…However, the patient’s liver function tests did not improve immediately. Approximately 2 weeks after admission, his serum was confirmed to be positive for both immunoglobulin A (IgA) anti-HEV and HEV RNA. IgG anti-HEV, IgM anti-HEV, and IgA anti-HEV were measured using an enzyme-linked immunosorbent assay (ELISA), and HEV RNA was measured via real-time RT-PCR, according to previously described methods 9–13]. We diagnosed him as having an HEV genotype 3b infection; as his hepatitis B virus (HBV) DNA levels were 1 log IU/mL (LIU/mL), the acute exacerbation or breakthrough of chronic hepatitis B were also excluded (Table 1). When he was discharged from our hospital 16 days after admission, his ALT and total bilirubin levels had improved (57 IU/L and 2.5 mg/dL, respectively), although he was still positive for HEV RNA (Figure 3). Notably, the titer of HEV RNA, determined via the previously described method [11], was found to be lower (3.8 x 102 copies/mL) on discharge than on days 5, 7, 12, and 14 (4.4 x 104 copies/mL, 2.4 x 104 copies/mL, 2.3 x 103 copies/mL, and 8.5 x 102 copies/mL, respectively). Other causes of viral hepatitis and autoimmune liver diseases were also ruled out. Drug lymphocyte stimulation tests (DLSTs) were also negative for flavoxate hydrochloride, vibegron, and silodosin [14]. Finally, we diagnosed him with an acute HEV infection after he was discharged.

In Discussion section, lines 129-138,

According to the World Health Organization (WHO), the incubation period following exposure to HEV ranges from 2 to 10 weeks (with mean ranges of 5 to 6 weeks), and HEV-infected persons excrete HEV beginning from a few days before to 3–4 weeks after the onset of the disease [17]. They also stated that chronic HEV genotype 3 or 4 infections had been reported in immunosuppressed people, particularly organ transplant recipients on immunosuppressive drugs. Unfortunately, we could not evaluate HEV RNA and ALT levels 2–3 months after the documented episode to confirm a negative HEV RNA result or further reduction in ALT levels, although, upon following up with this patient after 16 days, we measured a decrease in both HEV genotype 3b RNA levels and ALT levels with jaundice.

In Discussion section, lines 217-233,

…In the present case, hepatitis D virus (HDV) RNA was not measured. The prevalence of HDV infection is low in Japan, seen in 4 (1.2%) out of 323 HBsAg carriers in the Tokyo–Chiba area [91], although it was found in 32 (8.5%) of 375 HBsAg-positive patients in Miyako Island, Okinawa [9]. Although HBV DNA was well suppressed before, during and after the elevation of ALT levels, the involvement of HDV infection may not be completely ruled out. In general, HDV infection can be successfully blocked using a combination of hepatitis B immunoglobulin and a vaccine [9].

Recent recommendations [93] make the most sense in countries with a low prevalence of HBV and those with a high prevalence of HBV and HDV. The recently measured prevalence of HDV infection in patients with HBV infection in Hokkaido, Japan, was 1.7% [94]. Patients with serum positive for anti-HDV antibody experienced rapid progression of liver fibrosis, suggesting the importance of routine HDV testing. As the number of people migrating from abroad into Japan increases, the prevalence of HDV infection may increase. Further studies are needed at this point.

In Conclusions section, lines 242-247,

…We reported a male case with chronic hepatitis B and choledocholithiasis, who presented with a high elevation in ALT levels which was caused by HEV infection. It is important to consider other factors, including HEV infection, in patients with prolonged abnormal liver function after the removal of common bile duct stones. Further investigation of the role of HDV infection in the elevation of ALT levels is also needed.

Reply to your comment 2: “There is an abundance of references (probably a bit more than a case report needs, including the some 10% of the papers where one of the authors of this manuscript has been listed), most of those used in the Discussion section. I advise the authors to rewrite the discussion part, putting the emphasis on their case, other possible causes of the elevated aminotransferase and then comparing it to other similar case reports.”

Thank you very much for your invaluable comments. We need these important references and 1st reviewer recommended references of methods. Please see “Response to the comments from Reviewer 1”.

Round 2

Reviewer 1 Report

Comments and Suggestions for Authors

The authors partially responded to the concerns of the reviewer. A bracket is missing in line 103.

Author Response

Thank you for your invaluavle comment. According to your suggestion, we revised our manuscript with yellow highlight.